

# The development of Negative Self-Beliefs Inventory (NSBI): cultural adaptation and psychometric validation

Xiaoqing Tang[1],*, Wenjie Duan[1],*, Ying Wang[2] and Pengfei Guo[3]

[1] Department of Applied Social Sciences, City University of Hong Kong, Hong Kong, China
[2] Horizon Research Consultancy Group, Beijing, China
[3] Hospital (T.C.M.) Affiliated to Sichuan Medical University, Luzhou, Sichuan, China
* These authors contributed equally to this work.

## ABSTRACT

Social anxiety is an emotional disorder common to various populations around the world. The newly developed Self-Beliefs Related to Social Anxiety Scale (SBSA) aims to assess three kinds of self-beliefs through 15 items that include self-related cognitive factors that evidently result in social anxiety. This study explored the psychometric characteristics of SBSA among 978 Chinese. An eight-item Negative Self-beliefs Inventory (NSBI) was developed through qualitative and quantitative analyses. Exploratory factor analysis, confirmatory factor analysis, and multi-group confirmatory factor analysis suggested that NSBI contained clear, meaningful, stable, and invariant three-factor structure consistent with the original SBSA. Further analyses showed that the three subscales and the entire scale exhibited high internal consistency (0.779–0.837), good criterion validity, and good convergent and divergent validity (i.e., negative associations with flourishing and positive associations with anxiety, depression, and stress). These findings indicated that NSBI is reliable and valid for measuring negative self-beliefs in the Chinese population. A higher total score of NSBI indicates the more serious negative self-beliefs. Limitations of the present study and implications for research and practice were also discussed. Further studies are needed to evaluate the predictive ability, incremental validity, and potential role of NSBI in clinical and large-scale populations.

## INTRODUCTION

Mild anxiety or discomfort experienced by individuals when speaking in public or social situations is a normal psychological reaction. However, when the anxiety or discomfort causes severe distress and impairs normal social functioning, it may evolve into a mood disorder called social anxiety disorder. Social anxiety is one of the major emotional disorders that is characterized by remarkable and persistent fear of negative evaluation in social related contexts (*American Psychiatric Association, 2013*). In Western countries, the lifetime prevalence of social anxiety disorder was estimated at 12.10%–13.00% among different adult populations (*Furmark, 2002*; *Kessler et al., 2005*; *Polo et al., 2011*). Very few studies have investigated the prevalence of social anxiety disorder among the general

Corresponding author
Pengfei Guo, lyzyydb@163.com

Chinese adult population (*Hofmann, Asnaani & Hinton, 2010*). A recent large-scale research revealed that the 12-month and lifetime prevalence of social anxiety disorder among 11,527 Chinese military personnel were 3.34% and 6.22%, respectively (*Wang et al., 2014*). The low prevalence in China, as with other psychiatric disorders (e.g., depression, *Smith, 2014*), may be partly attributed to culture-related low detection rate (*Hofmann, Asnaani & Hinton, 2010*; *Smith, 2014*).

*Hofmann, Asnaani & Hinton (2010)* examined the cultural factors related to social anxiety and concluded that the degree of expression of social anxiety depends on the social norms, cultural background, and ethnic/racial characteristics. Accordingly, culture-related factors need to be carefully considered in different countries when conducting social anxiety-related assessment and treatment (*Hofmann, Asnaani & Hinton, 2010*). For instance, *Caldwell-Harris & Ayçiçegi (2006)* investigated the effect of individualism-and-collectivism on the reported psychological distress in both individualism and collectivism countries. The results indicated that the respondents living in collectivism countries usually reported low symptoms on anxiety, depression, schizophrenia, and antisocial personality disorder (*Caldwell-Harris & Ayçiçegi, 2006*). Another study further found that the participants in collectivistic cultures showed higher social anxiety levels and more positive attitude to socially avoidant behaviors. These findings implied that it is important and meaningful to conduct cultural adaptation when applying western (i.e., individualism countries) social anxiety inventories into eastern countries (i.e., collectivism countries).

## Self-beliefs of social anxiety

Determining the risk factors of social anxiety is very important in developing intervention programs and psychotherapies. *Ng, Abbott & Hunt (2014)* conducted a systematic review of 17 evidence-based studies and identified that negatively perceived self-related information (e.g., negative self-imagery) is the key cognitive factor that increases social anxiety in both clinical and non-clinical populations, as it had been emphasized in different cognitive models of social anxiety (*Clark & Wells, 1995*; *Hofmann, 2007*; *Rapee & Heimberg, 1997*). For instance, *Clark & Wells (1995)* stated that individuals' excessive attention to internal negative thoughts, feelings, and physical sensations in social contexts would confirm their perceived negative impression and beliefs of themselves that, in turn, would increase the level of anxiety. Similarly, *Rapee & Heimberg (1997)* and *Hofmann (2007)* recognized that social anxiety results from a discrepancy between individuals' negatively perceived self-related information and the assumed audiences' high expectation. Therefore, these cognitive models of social anxiety and cognitive–behavior therapies suggest that a reduction of negative self-related beliefs would positively relieve social anxiety.

Based on the importance of cognitive factors in social anxiety, *Wong & Moulds (2009)* developed the Self-Beliefs Related to Social Anxiety Scale (SBSA), which measures three types of self-beliefs in social contexts proposed by *Clark & Wells (1995)*. The scale consists of 15 items (i.e., four items for high standard beliefs on social performance, HSB; seven items for conditional beliefs on social evaluation, CB; and four items for unconditional beliefs on the self, UB). Preliminary psychometric evaluation demonstrated that the scale

displays excellent reliability (i.e., Cronbach's alpha >.82), satisfactory item–item and item–total correlations (i.e., Pearson correlations ranged from .72 to .89), meaningful factor structure, good convergent and divergent validity, and acceptable incremental and discriminative validity (*Wong & Moulds, 2009*; *Wong & Moulds, 2011*; *Wong, Moulds & Rapee, 2014*). Nevertheless, the stability of the factor structure was unclear. *Wong & Moulds (2011)* revealed a two-factor structure (i.e., CB and UB merged into one factor, and HSB was the other factor) in exploratory factor analysis using 600 non-clinical undergraduates, whereas the following confirmatory factor analysis (CFA) demonstrated that the three-factor structure exhibited better fit than the two-factor model. Finally, they adopted the three-factor model (*Wong & Moulds, 2011*) that is consistent with the *Clark & Wells (1995)*'s theoretical model. Recently, *Heeren et al. (2014)* likewise examined the structural validity of SBSA among a French-speaking community sample. Their study utilized CFA and revealed its replicable three-factor structure, good reliability, and concurrent validity.

It should be noted that social anxiety-related assessment is culturally dependent (*Hofmann, Asnaani & Hinton, 2010*). Equivalence of concepts and inventory items should be evaluated and adjusted before the western culture-based measurement can be applied into eastern countries.

## Equivalence of inventories in different cultures

Inconsistency in the obtained factor structures may be attributed to different interpretations of the items, which were referred to as functional equivalence and conceptual equivalence of items in previous studies (*Cheung, Van de Vijver & Leong, 2011*; *Duan et al., 2012*). The aforementioned studies were also conducted in Western countries. No study yet has examined the factor structure and psychometric characteristics of SBSA in Eastern cultures, hence there is a need to examine the cognitive understanding of each item in the context of Chinese culture and to re-explore and validate the factor structure.

Previous studies suggested that cultural adaptation should be considered to ensure equivalence of inventories in different cultures (*Ho et al., 2014b*). Specifically, *Johnson (1998)* proposed that cross-cultural equivalence of inventories should be obtained through four kinds of equivalences, namely, linguistic, conceptual, metric, and functional equivalence. Linguistic equivalence refers to the linguistic accuracy of each item in different cultures and emphasizes quality of translation. Conceptual equivalence refers to similarity in participants' understanding of factors and concepts despite coming from different cultures. Metric equivalence avoids the floor and/or ceiling effects. Finally, functional equivalence indicates that the behavior and/or thoughts described by the items are the same in different cultural contexts (*Ho, Duan & Tang, 2014a*; *Ho et al., 2014b*; *Johnson, 1998*).

Traditionally, translation and back-translation, as well as confirmatory factor analysis have been recognized as the most commonly used approaches in cross-cultural psychometric evaluation studies. Nevertheless, several scholars (e.g., *Hui & Triandis, 1985*; *Kankaraš & Moors, 2010*) argued that only part linguistic and conceptual equivalence could be obtained through the aforementioned traditional steps, and that the equality of translations, cultural relevance, measurement equivalent of constructs, and validity of the

adapted instrument need to be additionally and carefully considered. *The World Healh Organization (2011)* published a four-step guideline for refining the original "translation and back-translation" method, which emphasized the role of experts in moderating the equality of translations and partly composed the deficiencies of traditional approach. Metric and functional equivalences can often be explored through qualitative methods, such as group interview. However, very few researchers have done so. Our previous experience demonstrated that the combination of these rules and methods is helpful in ensuring the equivalence of measurement tools in different cultures. For instance, Values in Action Inventory of Strengths (VIA-IS) is a widely used measurement for assessing character strengths among diverse populations by using 240 items (*Peterson & Seligman, 2004*). However, the factor structures of the VIA-IS are various in different countries (*Duan et al., 2012*). Further qualitative and quantitative inspections found that several items were common social expectations that may lack sensitivity; several items may not be appropriate in representing the spirituality-related culture in Mainland China; and several items may represent socially unacceptable behavior in Mainland China (cf. *Duan et al., 2012*; *Ho, Duan & Tang, 2014a*). After these inappropriate items have been deleted, a culturally fit and stable strength-structure was obtained (*Duan et al., 2013*).

## The present study

We expect to achieve several objectives, namely to (1) obtain equality between the English and Chinese translations and examine the cultural relevance of each item through qualitative and quantitative analyses; (2) validate the Chinese version of SBSA and test its factor structure, reliability, criterion validity, convergent validity, and divergent validity; specifically, SBSA should show positive relations with trait anxiety, state anxiety, social anxiety, stress, and depression; whereas show negative relations with psychological wellbeing (e.g., Flourishing); (3) establish the cross-gender measurement invariance for meaningful comparisons between different groups, which can guarantee similar latent constructs across groups (*Vandenberg & Lance, 2000*); and (4) to obtain solid psychometric evidence through a short form that is practical and convenient to apply in the community, clinical, and large-scale settings for purposes of research and intervention evaluation (*Ziegler, Kemper & Kruyen, 2014*).

## METHOD

### Participants and procedures

A total of 978 (428 males, 550 females; Mean age $= 20.73$, $SD = 3.46$) participants from six different universities were involved in this quantitative survey. Those universities are located in Eastern, Central, and Western China and this distribution is helpful in balancing the economic and social background of the participants. Participants with active physical and mental illnesses were excluded. No participant reported serious medical history or long-term medication. The participants were asked to provide written informed consent before completing the questionnaires. The Institutional Review Board of the Southwest University approved this study.

The entire sample was divided into four independent subsamples; each subsample completed a distinct questionnaire package created for specific research purposes to control the source of common-method bias (*Podsakoff et al., 2003*; *Podsakoff, MacKenzie & Podsakoff, 2012*) and reduce participants' cognitive load and fatigue (*Rammstedt & Beierlein, 2014*). Subsample 1 ($n_1 = 330$; 171 males, 159 females; Mean age = 20.42, *SD* = 0.77) completed the Chinese version of SBSA for exploratory factor analysis; subsample 2 ($n_2 = 330$; 164 males, 166 females; Mean age = 20.40, *SD* = 0.73) also completed the Chinese version of SBSA but for confirmatory factor analysis; subsample 3 ($n_3 = 155$; 44 males, 111 females; Mean age = 21.45, *SD* = 6.12) completed the short form of SBSA, Liebowitz Social Anxiety Scale, and State-Trait Anxiety Inventory for examining criterion validities; and subsample 4 ($n_4 = 163$; 49 males, 114 females; Mean age = 21.38, *SD* = 0.73) completed the short form of SBSA, Depression Anxiety Stress Scales, and Flourishing Scale for examining the convergent and divergent validities.

After the investigation was completed, the study objective and corresponding interpretations were explained to the participants. Data were collected from May to November in 2014.

## Measurements

### Self-Beliefs Related to Social Anxiety (SBSA)

SBSA is a 15-item self-reporting questionnaire that assesses the strengths of self-perceived beliefs related to the self in social contexts (*Wong & Moulds, 2009*; *Wong & Moulds, 2011*; *Wong, Moulds & Rapee, 2014*). It contains three subscales (four-item HSB, seven-item CB, and four-item UB). Participants were asked to rate each item on an 11-point Likert scale from 0 (do not agree at all) to 10 (strongly agree). Subscale scores and total scores were calculated by summing up the scores of the corresponding items. High scores reflect the strong strengths of self-beliefs.

### Liebowitz Social Anxiety Scale (LSAS)

LSAS is a 24-item self-reporting scale that measures anxiety and the avoidance of various social performances and situations (*Liebowitz, 1987*). For each social performance and situation, participants were required to rate their feelings and behaviors on a four-point Likert scale ranging from 0 (never) to 3 (always). High scores of the total scale indicate increased levels of social anxiety. The Chinese version of LSAS processed good psychometric properties among both clinical and non-clinical populations (*He & Zhang, 2004*). The Cronbach's alpha of the current sample is .934.

### State-Trait Anxiety Inventory (STAI)

STAI is a widely used self-reporting inventory for assessing the state (20 items) and trait (20 items) of anxiety among diverse populations (*Spielberger, Gorsuch & Lushene, 1970*). Different instructions for the two subscales were provided to guide participants in giving appropriate responses. All items were rated on a four-point Likert scale ranging from 0 (never) to 3 (very obvious/always). The scores of two subscales were summed separately; high scores reflected increased levels of state anxiety or trait anxiety. The Cronbach's alpha of the state and trait subscales in the current sample are .893 and .847, respectively.

### Depression Anxiety Stress Scales (DASS)

Depression, anxiety, and stress over the past week were assessed through a short version of DASS, which is a 21-item self-reporting scale that contains three subscales (seven items per subscale) (*Lovibond & Lovibond, 1995*). Previous studies revealed its good internal consistency and factor structure (e.g., *Antony et al., 1998*). High scores of the three subscales separately reflect high level or severity of depression or anxiety. The Cronbach's alpha of the current study is .859.

### Flourishing Scale (FS)

FS is a new inventory that assesses the important aspects of the functioning of human functioning through eight items (*Diener et al., 2010*), which reflects the general psychological wellbeing of individuals. Participants used a seven-point Likert scale to evaluate the items by using 1 (strongly disagree) to 7 (strongly agree). A high mean score of the whole scale indicates a high degree of psychological wellbeing. *Tang et al. (2014)* demonstrated its good psychometric characteristics among the Chinese. The Cronbach's alpha of the current sample is .789.

## Translation of SBSA

The steps recommended by *Hambleton, Merenda & Spielberger (2004)* and *Sperber (2004)* are comprehensively considered to achieve linguistic equivalence in the present study. The first author of this manuscript established a triangular group, including one PhD student majoring in Psychology, one PhD student majoring in Sociology, and one psychology professor who severed as moderator. All members are bilingual experts (i.e., English and Chinese). First, the original SBSA was translated into Chinese by the psychology PhD student. The sociology PhD student then back-translated the Chinese version of SBSA into English. The professor supervised the entire translation process and was responsible for verifying the conformity of the translated English items with the original ones, as well as the precision of the Chinese items. Discrepancies were discussed thoroughly and revised by the triangular group and the first author.

## Data analysis plan

Data analysis was composed of both qualitative and quantitative stages.

The qualitative stage aimed to conduct cognitive interview among undergraduates to obtain feedback regarding the appropriateness and meanings of the SBSA items in the context of Chinese culture. Previous studies (e.g., *Duan et al., 2012*) suggested that cognitive feedback from the target group would be helpful in refining the translations and/or determining what culturally inappropriate items to delete. The first author conducted interviews among 20 undergraduates who were unaware of the purpose of the study and had not attended the quantitative survey. Four types of standardized questions, which were used in previous studies, were presented to them (*Duan et al., 2012*): (1) Please tell me whether you understand this item or not. What do you think the item is asking? (2) What did you think about when you first read this item? (3) Do you understand the description of response choices in the questionnaire? What is the meaning of "strongly

agree"? (And so on for other responses) Which one do you choose? Why? (4) Could you select a response choice that reflects your true opinion of this item? Why? Questions (1) and (2) assessed conceptual and functional equivalence, whereas Questions (3) and (4) assessed metric equivalence.

During the quantitative analysis, the first step was to calculate for internal consistency (i.e., Cronbach's alpha) by using the first subsample. Items that could improve the internal consistency coefficient when deleted were considered to be removed. As what the original authors did (*Wong & Moulds, 2009*), maximum likelihood factor analysis with promax rotation method was adopted to evaluate the factor structure. Confirmatory factor analysis and multi-group confirmatory factor analysis were conducted using the second subsample to identify the best-fit model and evaluate measurement invariance across genders. A short form with high factor loadings, clear factor structure, and measurement invariance across different gender groups was expected to be developed based on the above steps.

Criterion validity, convergent validity, and divergent validity of the short form were further tested using the third and fourth subsamples. Pearson correlations were calculated between the short form and similar psychological variables (e.g., trait anxiety, state anxiety, and social anxiety), psychological distress (e.g., depression, stress, and anxiety), and psychological wellbeing (e.g., flourishing).

Data were analyzed using SPSS 21.0 and Mplus 7.0.

## RESULTS

### Cognitive interview

The results of cognitive feedback revealed that several items in the 15-item SBSA might contain conceptual and functional issues, but no metric issue was proposed. Most participants ($n = 18$) indicated that Item 4 "I have to appear intelligent and witty" and Item 7 "I have to convey a favorable impression" described strategies of impression management in Chinese culture rather than self-beliefs because the information conveyed by the items met Chinese social expectations. In other words, individuals in front of other people are always prone to impress and show their good side. Accordingly, Item 4 and Item 7 are "positive behaviors" and "advisable beliefs", rather than the negative beliefs related to anxiety in Mainland China. Most of the participants would rate highly on the two items. In addition, more than half of the respondents ($n = 12$) considered Item 3 "If people do not accept me, I'm worthless" and Item 5 "If someone does not like me, it must be my fault" possibly refers to high-standard self-beliefs, which indicate that an individual should be valuable and get people to like him/her. In other words, Items 3 and 5 may be conceptually varied in Western and Eastern societies. Additionally, several participants ($n = 9$) thought that Item 1 "If I make mistakes, others will reject me" had uncertain meaning, especially with regard to the word of "mistake". The severity of the mistakes would affect their rating of this item. For instance, some students said that if the mistake was really small or only related to him/her selves, then others would not reject them; on the other hand, if the mistake really mattered or impaired damaged the collective interest, then others would reject them. Thus the "mistake" may have different meanings to different individuals.

Several students ($n = 14$) did not understand why anxiety would be a sign of weakness (Item 8 "If people know I'm anxious, they will think I'm weak."). Accordingly, we assumed that these items (i.e., Items 1, 3, 5, 4, 7, and 8) might lack sensitivity of assessment in Chinese culture, and that their removal will improve the scale's reliability and validity. Nevertheless, it should be noted that the items are not directly removed based on the above cognitive interview results. Both qualitative and quantitative results should be considered before the removing of the items.

## Internal consistency

The Cronbach's alpha of the original 15-item scale was 0.880. However, the results suggested that the alpha would increase to .888 if Item 4 was deleted. After Item 4 has been removed, the results again suggested that the removal of Item 7 would increase the Cronbach's alpha to .891. Integrating the results of cognitive interviews, Items 4 and 7 were removed from the 15-item pool.

## Exploratory factor analysis

Maximum likelihood factor analysis with promax rotation method was performed among the remaining 13 items. KMO = .895 and Bartlett's Test of Sphericity = 1978.376 ($p < .001$) indicated that the current data pool was appropriate for analysis. Three factors were extracted, but several items were cross-loaded. For instance, Item 1 was loaded on factors 2 (loading = .487) and 3 (loading = .465); thus, Item 1 was removed. After several explorations, Items 2, 3, 5, and 8 were removed as cross-loadings. The removed items were likewise questionable, as reflected by the cognitive interviews. Finally, eight items were left (i.e., items 6, 9, 10, 11, 12, 13, 14, and 15) for the final factor analysis. The result indicated that the eight-item pool remained appropriate for factor analysis (KMO = .851; Bartlett's Test of Sphericity = 1263.272, $p < .001$), and a clear three-factor structure was obtained (Table 1). All factor loadings of the eight items were higher than .67. Considering the content validity of the revised inventor with regards to the original scale, the shortened scale was renamed as the Negative Self-beliefs Inventory (NSBI). The Cronbach's alpha of the HSB subscale in the NSBI was .779, that of the CB subscale was .784, and that of the UB subscale was .837. These results indicated that the internal consistency of the NSBI was good (*Maydeu-Olivares, Coffman & Hartmann, 2007*).

## Confirmatory factor analysis

Subsample 2 ($n_2 = 330$) was used to further investigate the factor structure of NSBI through confirmatory factor analysis. Comparative Fit Index (CFI >.95), and Root-mean-square Error of Approximation (RMSEA <.50 or .80) were adopted to evaluate the comparable models and/or structural equation models (*Hu & Bentler, 1999*). Following three previous studies (*Heeren et al., 2014*; *Wong & Moulds, 2009*; *Wong & Moulds, 2011*; *Wong, Moulds & Rapee, 2014*), three comparable models were constructed, including a three-factor model, a two-factor model (i.e., the items of CB subscale and UB subscale loaded on the same factor), and a single-factor model (i.e., all items loaded on one factor). The goodness-of-fit indices of the three models are shown in Table 2 and suggest that the

Table 1 **Maximum likelihood factor analysis of the Negative Self-Beliefs Inventory.** ($n_1 = 330$).

| | Items | Factor | | |
| | | UB | CB | HSB |
|---|---|---|---|---|
| Item 9 | 人们认为我是差劲的 (People think I'm inferior) | .856 | | |
| Item 14 | 人们不尊重我 (People don't respect me) | .784 | | |
| Item 6 | 人们认为我很糟糕 (People think badly of me) | .765 | | |
| Item 12 | 如果人们看到我焦虑，他们会对我失望 (If people see me anxious, they'll put me down) | | .813 | |
| Item 13 | 如我不说一些有趣的事情，人们就不会喜欢我 (If I don't say something interesting, people won't like me) | | .708 | |
| Item 10 | 如果我不把所有事情弄好，我就会受到排斥 (If I don't get everything right, I'll be rejected) | | .694 | |
| Item 15 | 我需要被所有人喜欢 (I need to be liked by everyone) | | | .995 |
| Item 11 | 我必须得到所有人的认可 (I must get everyone's approval) | | | .673 |
| | % of variance | 26.88% | 33.76% | 5.77% |

**Notes.**

UB, Unconditional beliefs about self; CB, Conditional beliefs concerning social evaluation; HSB, High standards for social performance.

Table 2 **Goodness-of-fit indices for confirmatory factor analysis.** ($n_2 = 330$).

| | Goodness-of-fit indices | | | |
| | CFI | TLI | RMSEA | 90% CI |
|---|---|---|---|---|
| Three-factor model | .961 | .935 | .079 | [.055, .104] |
| Two-factor model | .854 | .785 | .171 | [.150, .193] |
| Single factor model | .740 | .636 | .222 | [.202, 243] |

three-factor model achieved the best fit in our sample. Standardized path coefficients of the three-factor model are shown in Fig. 1 and are significant at .001 levels. All standardized item loadings were higher than .710. These results supported the three-factor structure of NSBI among the Chinese undergraduate population.

## Multi-group confirmatory factor analysis

*Meredith (1993)* and *Byrne (2012)* recommended that measurement invariance could be achieved by examining the four levels of equivalence from the weakest to the strongest, including configural invariance, weak/metric factorial invariance, strong/scalar factorial invariance, and the variance of the means of latent variables. Two criteria were used to determine whether equivalence was maintained between a more restricted model and a less restricted one, including the change in CFI ($\Delta$CFI) and change in RMSEA

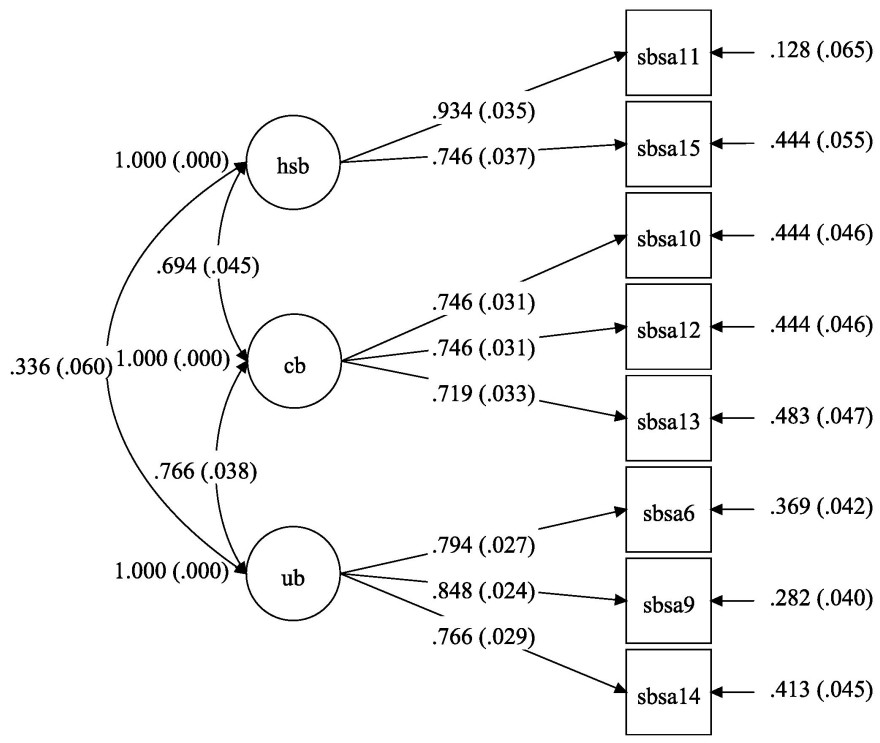

**Figure 1 Confirmatory factor analysis model of the Negative Self-Beliefs Inventory with standardized path coefficients.** ub, Unconditional beliefs about self; cb, Conditional beliefs concerning social evaluation; hsb, High standards for social performance.

**Table 3 Invariance test across gender of NSBI. ($n_2 = 330$).**

|  | $\chi^2$ | $df$ | CFI | $|\Delta\text{CFI}|$ | RMSEA | $|\Delta\text{RMSEA}|$ |
|---|---|---|---|---|---|---|
| Gender group |  |  |  |  |  |  |
| Model one | 64.958 | 34 | .965 | – | .074 | – |
| Model two | 72.588 | 39 | .962 | .003 | .072 | .002 |
| Model three | 81.425 | 44 | .958 | .004 | .072 | .000 |
| Model four | 93.211 | 47 | .948 | .010 | .077 | .005 |

**Notes.**

Model one, configural model; Model two, equal loadings model; Model three, equal loadings + intercepts model; Model four, equal loadings + intercepts + means model.

($\Delta$RMSEA). Researchers suggested that $|\Delta\text{CFI}| < .010$ (*Cheung & Rensvold, 2002*) and $|\Delta\text{RMSEA}| < .015$ (*Chen, 2007*) supported the equivalence of measurement. *Chen (2007)* considered $|\Delta\text{RMSEA}|$ as an important supplement indicator when the total sample size was larger than 300, as with the current one ($n_2 = 330$). The results presented in Table 3 reveal acceptable changes in the CFI and RMSEA, which supported the measurement equivalence of NSBI in the different gender groups.

Table 4 **Pearson correlations between the NSBI and other anxiety related scales.** ($n_3 = 155$).

|  | HSB | CB | UB | NSBI |
|---|---|---|---|---|
| State anxiety | .129 | .212[**] | .336[**] | .269[**] |
| Trait anxiety | .169[*] | .196[*] | .357[**] | .286[**] |
| LSAS | .052 | .198[*] | .220[**] | .189[*] |

**Notes.**
UB, Unconditional beliefs about self; CB, Conditional beliefs concerning social evaluation; HSB, High standards for social performance; LSAS, Liebowitz social anxiety scale.

[*] $p < .05$.

[**] $p < .01$.

Table 5 **Pearson correlations between the NSBI and psychological outcomes.** ($n_4 = 163$).

|  | HSB | CB | UB | NSBI |
|---|---|---|---|---|
| Flourishing | −.108 | −.239[**] | −.303[**] | −.261[**] |
| Anxiety | .272[**] | .235[**] | .273[**] | .308[**] |
| Depression | .241[**] | .299[**] | .296[**] | .334[**] |
| Stress | .300[**] | .384[**] | .348[**] | .414[**] |

**Notes.**
UB, Unconditional beliefs about self; CB, Conditional beliefs concerning social evaluation; HSB, High standards for social performance.

[**] $p < .01$.

### Criterion validity

Criterion validity was examined using the third subsample ($n_3 = 155$). Pearson correlation results are shown in Table 4. In addition to the HSB subscale, the CB subscale, UB subscale, and total scale of NSBI were positively related ($r = .160–.357$) to other anxiety-related measurements, including state anxiety, trait anxiety, and LSAS. Among the three subscales of NSBI, the UB subscale displayed the highest correlation coefficients.

### Convergent and divergent validity

Convergent and divergent validities were examined by calculating the Pearson correlations between NSBI and both the negative and positive psychological outcomes. As expected, all subscales and the entire scale exhibited negative relations with flourishing ($r = −.108$ to $−.303$) and positive relations with depression, anxiety, and stress ($r = .235–.414$) (Table 5). All correlation coefficients were significant at .001 levels with the exception of the HSB subscale.

## DISCUSSION

The aim of this study was to validate the culturally adapted SBSA. Through a series of statistical analysis, an eight-item NSBI was developed and was proven to be capable of providing stable and clear three-factor structure, acceptable reliability, good criterion, convergent, and divergent validity.

A total of seven items (items 1, 2, 3, 4, 5, 7, and 8) were removed from the original 15-item pool (*Wong & Moulds, 2009*) through cognitive interview and exploratory factor

analyses. Previous related studies also found that several of these items were questionable. For instance, in the deleted items, *Heeren et al. (2014)* indicated that the item loadings of Items 1, 4, and 7 in the French version were lower than .40. Similarly, Items 4 and 7 exhibited the lowest item loadings among all four items of the HSB subscale, and Item 2 was the lowest among all items of UB subscale (*Wong, Moulds & Rapee, 2014*). The removal of these low loading items improved the factor structure to some extent. Actually, both the removing and remaining items highlighted the role of self-construals in cross-cultural social anxiety studies, which defined how people relate to others and the social context (*Hofmann, Asnaani & Hinton, 2010*). The removed items were all related to independent self-construals (e.g., Item 4 "I have to appear intelligent and witty" and Item 7 "I have to convey a favorable impression"), which were frequent in western countries or individualistic societies and reflected the tendency of viewing self as autonomous from the social context; whereas the remaining items were all related to interdependent self-construals (e.g., Item 6 "People think badly of me" and Item 10 "If I don't get everything right, I'll be rejected"), which were common in eastern countries or collectivist societies and reflected the tendency of viewing self as being integrated with others and social context (*Hofmann, Asnaani & Hinton, 2010*).

In addition, as discussed previously, a two-factor structure from exploratory factor analysis was against the three-factor structure from confirmatory factor analysis (*Wong & Moulds, 2011*). After deleting several cross-loading items, the three-factor structure was clearly obtained through exploratory factor analysis and further validated through exploratory factor analysis (using another independent sample). Thus, the results supported the possibility of the cross-loading phenomenon as the cause of inconsistency in the results of *Wong & Moulds (2011)* and *Wong & Moulds (2009)*. Reports from the revised eight-item NSBI also preliminarily revealed measurement equivalence across gender groups. The overall fit of the four levels of invariance models was acceptable, which means that indicators (i.e., items) load on similar factors with equal factor loadings across different groups (*Bontempo & Hofer, 2007*), and that the corresponding factor intercepts and latent mean differences were equivalent across genders. Thus, meaningful comparisons of the three factors of NSBI can be made in different gender groups (*Vandenberg & Lance, 2000*).

NSBI was expected to exhibit high correlation with LASA because the SBSA-CS reflected social anxiety-related beliefs and a low correlation with state and trait anxiety. However, the current study obtained unexpected results; SBSA-SF had a high correlation with general anxiety (i.e., state and trait anxiety), and relative low correlation with LASA. This result was likewise found in a French-speaking sample (*Heeren et al., 2014*). We considered the lack of a clinical sample of social anxiety as the reason behind the above results because this study recruited college students who may not exhibit high scores on social anxiety as participants. In terms of relationship between NSBI and state/trait anxiety, UB subscale exhibited the highest correlations with state/trait anxiety among the three subscales. The negative evaluation reflected by the UB subscale was recognized as a trait-like vulnerability (*Chase et al., 2010*; *Clark, 2002*; *Weeks & Howell, 2012*), which was associated with a wide

range of emotional disorders (e.g., general anxiety, social anxiety, and depression). This association also explains why NSBI was associated with depression in the present study as well as in previous ones (*Heeren et al., 2014*; *Wong, Moulds & Rapee, 2014*). NSBI was also negatively related to flourishing and positively related to stress. All of these results indicate acceptable convergent and discriminant validities.

Many studies have found similar cognitive patterns and co-occurrences between individuals with social phobia and depressive disorders (*Dozois & Frewen, 2006*; *Wittchen & Fehm, 2001*). This observation could be another possible explanation as to why NSBI was associated with anxiety and depression. Numerous studies demonstrated that rumination was a cognitive trigger of depression, and reduced rumination thinking had a positive effect on depressive symptoms (e.g., *Smith & Alloy, 2009*; *Zawadzki, Graham & Gerin, 2013*). If the self-beliefs assessed by NSBI were important cognitive factors of social anxiety, and if these self-beliefs differed with rumination in conceptual and functional levels, then we can further hypothesize that rumination, compared to self-beliefs related to social anxiety, occupies incremental validity when predicting depression, and that compared to ruminations, self-beliefs related to social anxiety occupy incremental validity when predicting social anxiety. Verifying the above hypotheses and clarifying the relationship between rumination and self-beliefs require further examination through a longitudinal research design and clinical samples in the future.

Several limitations of this study should be identified. The major limitation of this study is the use of the university student sample and lack of clinical participants. Hence, this sampling limits the generalizability of the results to all Chinese adults. Furthermore, some of the items being removed in the cognitive debriefing may indicate their immaturity about the social norm and society expectation in Chinese context. Future studies should re-examine the reliability and validity of NSBI with a clinical sample of people suffering from social anxiety disorder and/or a community sample. Second, limited validities were examined in the current study. Future studies should examine whether NSBI exhibits incremental validities when compared to other factors (e.g., rumination) in predicting social anxiety. Third, longitudinal studies should be conducted to clarify the mediation role of NSBI before a meaningful intervention program can be developed. Finally, the short version of SBSA was obtained among the Chinese population. Although the short form of the scale was effective and timesaving in large-scale social surveys (*Rammstedt & Beierlein, 2014*), psychometric evaluations in other countries, especially in Western countries, should be evaluated further.

Our findings indicate that an eight-item Negative Self-beliefs Inventory (NSBI) provides reliable and valid observations on three kinds of maladaptive self-beliefs (*Clark & Wells, 1995*). According to the above findings, NSBI is related to psychological distress including depression, anxiety, and stress. A higher total score of NSBI reflects more serious negative self-beliefs, which in turn associates with higher level of psychological distress.

### Funding

The authors received no funding for this work.

### Competing Interests

The authors declare there are no competing interests.

### Author Contributions

- Xiaoqing Tang conceived and designed the experiments, analyzed the data, contributed reagents/materials/analysis tools, wrote the paper, prepared figures and/or tables, reviewed drafts of the paper.
- Wenjie Duan conceived and designed the experiments, performed the experiments, analyzed the data, contributed reagents/materials/analysis tools, wrote the paper, prepared figures and/or tables, reviewed drafts of the paper.
- Ying Wang conceived and designed the experiments, contributed reagents/materials/analysis tools, data Collection.
- Pengfei Guo conceived and designed the experiments, performed the experiments, contributed reagents/materials/analysis tools, data Collection.

### Human Ethics

The following information was supplied relating to ethical approvals (i.e., approving body and any reference numbers):

The participants were asked to provide written informed consent before completing the questionnaires. The Institutional Review Board of the Southwest University approved this study.

### Supplemental Information

Supplemental information for this article can be found online at http://dx.doi.org/10.7717/peerj.1312#supplemental-information.

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
