# Peer review of "The development of Negative Self-Beliefs Inventory (NSBI): cultural adaptation and psychometric validation"

_PeerJ, doi:10.7717/peerj.1312_

## Round 0.1 · original submission · Minor Revisions

Many thanks for submitting your paper to PeerJ for consideration. This is a well written paper. However, before I could consider accepting it, there are a number of issues which I and two further expert reviewers have identified which need attention. The full reviews are attached. However, they each raise separate but important points. Both agree on the basic methodological soundness, but thereafter do have some difficulties. Reviewer one identifies issues with the referencing, evidence provided, and citations within the manuscript and makes some suggestions. In particular, I would like you to consider the extent of self-citation in this paper, and consider whether you have chosen the most appropriate reference from the wider literature in each case to cite.

In terms of the writing, I agree that the moderation function of the role of the flourishing variable needs greater clarity in the introduction.

Both reviewers have issues with the removal of several items - reviewer one in terms of the statistical implications, and reviewer two in terms of the content validity of the resultant scale. I agree with both of these points, and believe that they need to be addressed separately. Reviewer two raises an issue around the extent to which the scale is applicable to the general population or those with an identified disorder, and the ability of those in the cognitive interviews to give helpful feedback if they were not anxious participants, and these points do require some explanation. I believe that you should give serious thought to the final suggestion by reviewer two, to consider recasting this study as developing a new scale which would need evaluation for content validity.

Reviewer 1 ·

Basic reporting

This paper reports the psychometric characteristics of SBSA and validation of SBSA-SF which is an important tool for measuring social anxiety in Chinese populations. However, it is suggested to give the readers more ideas about the cultural differences on society anxiety issues in the introduction section to illustrate the research gaps of this study. Also, it is unclear and not well justified about the removal of items in developing the SBSA-SF. Moreover, the authors reported results and discussed the findings in the result section. The two sections should be very distinctive. In terms of research integrity, self-citation should be relevant and non-excessive. The authors cited many his papers which may not be necessary.

Experimental design

It is acceptable to use cross-sectional design. The sample size is strong enough for the study.

Specific comments:

Introduction -
Line 45: While the author is discussing the cultural difference of social anxiety, the following paper must be cited:
Hofmann, S. G., Asnaani, A., & Hinton, D. E. (2010). Cultural aspects in social anxiety and social anxiety disorder. Depression and anxiety, 27(12), 1117-1127.
Also, it is suggested to give more justifications on the difference on cultural difference on social anxiety to make the manuscript more unique and important to fill the research gaps. It is also worthy to explain the inadequacy of the Chinese version of Liebowitz Social Anxiety Scale (He & Zhang, 2004), which has been tested in clinical and non-clinical populations.
Line 71-75: These two sentences seem to be contradictory: “Wong & Moulds (2011) also found that items of CB and UB merged into one factor in exploratory factor analysis, whereas items of HSB were loaded on one separate factor. This finding contradicted the theoretical model. Confirmatory factor analysis (CFA) demonstrated that the three-factor structure exhibited better fit than the two-factor model (Wong & Moulds 2011).” Please clarify whether SBSA was confirmed as having 2- or 3-factor model in Wong & Moulds (2011)’s study. Also, what is the sample of that study? Non-Chinese population?

Line 109: The author explained the example of Values in Action Inventory of Strengths in detail in the introduction section, which may not be necessary. It is suggested to simply in to 2-3 short sentences.

Line 116 and 117: Is there any difference between “mainland” and “China”?

Line 119-121: The author cited his own article for many times. “Follow-up studies demonstrated the same positive functions of Chinese 120 virtues (e.g., Duan et al. 2012b; Duan & Guo 2015; Duan et al. 2015a; Duan et al. 2015b; Zhang121 et al. 2014b) compared to original ones proposed by Peterson & Seligman (2004).” Are they all necessary?

Line 126-127: Please cite reference for the sentence “Self-beliefs may constitute
one of the internal mechanisms that lead to social anxiety.”

Line 128: Please explain more about “virtues” and the reason why it is negatively related to anxiety.

Line 144: Why do the author needs a citation here?


Methodology-

Line 208: Why do the authors need to measure flourishing? Please give the definition and indicate the relationship between flourishing and social anxiety in the introductory section.

Line 214-5: “The steps recommended by Hambleton et al. (2004), Sperber (2004), and Duan et al.
215 (2012a) are comprehensively considered to achieve linguistic equivalence in the present study.” Please cite the original author invent this method.

Line 243: I don’t think promax rotation method is developed by Wong & Moulds (2009). Please double check.

Line 244: “A clear three-factor structure was the expected outcome according to previous studies (Heeren et al. 2014; Wong & Moulds 2009; Wong & Moulds 2011; Wong et al. 2014).” This sentence is not needed. The authors did not depend on the previous study findings to evaluate the factor structure of C-SBSA.

Validity of the findings

Line 264 and line 267: It is very confused of citing references in the results section. Please don’t discuss the findings in results section and move them into the discussion section.

Line 276-8: The reasons of removing item 1,3,5,4,7, 8 are not strong enough. Do you mean that vague meaning and high-standard self-belief were the reasons of rejecting the items? In fact, vague meaning may be due to the poor translation. Cognitive debriefing is a way to ensure the translation from English to Chinese is culturally adaptable.
Also, if impression management is relevant to Chinese social expectation, it should be relevant to retain those items in the SBSA to ensure Chinese cultural sensitivity. Why this is also one of the reasons to reject the items 4 and 7?
It is suggested to include all the factor loadings for all items in table 1 and see whether the removed items had low factor loading before the item removal.

Line 287: Once again, promax rotation method is not developed by Wong & Moulds. Please clarify.

Line 425: The major limitation of this study is the use of university student sample, meaning that the findings cannot be generalized to all Chinese adults, in particular, some of the items being removed in the cognitive debriefing may indicate their immaturity about the social norm and society expectation in Chinese context.

·

Basic reporting

I applaud Dr. Wenjie Duan and Dr. Xiaoqing Tang for such a thorough review of the Self-Beliefs related to Social Anxiety scale (SBSA-SF). I found the article well written and very solid in terms of their selection of the psychometric tests and presentation of the results.

Experimental design

I would however like to offer that removing items from the instruments does not ensure cultural equivalence. In fact this compromises the content validity of the revised scale with regards to the original scale. As indicated by the authors, cultural adaptation goes beyond simple translation. But instead of questioning the translation, authors elected to remove the items rather than considering adapting them. In addition, when developing a scale content validity plays a major role and the comprehensiveness of the overarching concepts (i.e., anxiety disorder) needs to be balanced when removing items based on a few decimal changes on the Cronbach’s alpha or factor loading for instance.

It is also unclear why the scale was revised and the SF validated upon feedback from participants who might not even display the symptoms or be diagnosed with the disorder.

Validity of the findings

I would recommend positioning the SF as a new scale and indicate that it will be validated upon testing in the targeted population.

---

## Round 0.2 · accepted · Accept

The revision to this paper has addressed the key issues identified by the reviewers. I am happy to now recommend this paper for acceptance in PeerJ.

Reviewer 1 ·

Basic reporting

It is good that the citation problem has been addressed.

Experimental design

Sounds appropriate.

Validity of the findings

Well justified.

Additional comments

The paper is much better after revision. No further comments.